# The Occluded Epitope Residing in Spike Receptor-Binding Motif Is Essential for Cross-Neutralization of SARS-CoV-2 Delta Variant

**Weeraya Thongkum [1,2,3], Kanyarat Thongheang [1,2,3] and Chatchai Tayapiwatana [1,2,3,*]**

[1]   Division of Clinical Immunology, Department of Medical Technology, Faculty of Associated Medical Sciences, Chiang Mai University, Chiang Mai 50200, Thailand; weeraya.t@cmu.ac.th (W.T.); kanyarat_thongheang@cmu.ac.th (K.T.)
[2]   Center of Biomolecular Therapy and Diagnostic, Faculty of Associated Medical Sciences, Chiang Mai University, Chiang Mai 50200, Thailand
[3]   Center of Innovative Immunodiagnostic Development, Faculty of Associated Medical Sciences, Chiang Mai University, Chiang Mai 50200, Thailand
[*]   Correspondence: chatchai.t@cmu.ac.th; Tel.: +66-81-8845141

**Abstract:** Concerns over vaccine efficacy after the emergence of the SARS-CoV-2 Delta variant prompted revisiting the vaccine design concepts. Monoclonal antibodies (mAbs) have been developed to identify the neutralizing epitopes on spike protein. It has been confirmed that the key amino acid residues in epitopes that induce the formation of neutralizing antibodies do not have to be on the receptor-binding domain (RBD)- angiotensin-converting enzyme 2 (ACE2) contact surface, and may be conformationally hidden. In addition, this epitope is tolerant to amino acid mutations of the Delta variant. The antibody titers against RBD in health care workers in Thailand receiving two doses of CoronaVac, followed by a booster dose of BNT162b2, were significantly increased. The neutralizing antibodies against the Delta variant suggest that the overall neutralizing antibody level against the Wuhan strain, using the NeutraLISA, was consistent with the levels of anti-RBD antibodies. However, individuals with moderate anti-RBD antibody responses have different levels of a unique antibody population competing with a cross-neutralizing mAb clone, 40591-MM43, determined by in-house competitive ELISA. Since 40591-MM43 mAb indicates cross-neutralizing activity against the Delta variant, this evidence implies that the efficiency of the vaccination regimen should be improved to facilitate cross-protective antibodies against Delta variant infections. The RBD epitope recognized by 40591-MM43 mAb is hidden in the close state.

**Keywords:** neutralizing epitopes; receptor-binding domain; receptor-binding motif

## 1. Introduction

Severe acute respiratory syndrome coronavirus 2 (SARS-CoV-2) causes coronavirus disease 2019 (COVID-19). It has infected more than 481.8 million people globally and caused over 6.1 million deaths [1–3]. Tremendous efforts have been put into discovering effective vaccines for preventing SARS-CoV-2 infection and disease transmission. SARS-CoV-2 shares high sequence homology with SARS-CoV and infects host cells via a similar mechanism [4,5]. These viruses use angiotensin-converting enzyme 2 (ACE2) as a cell receptor for viral entry through their transmembrane spike (S) glycoprotein. The S protein consists of two subunits, S1 and S2. The S1 subunit binds the target cells expressing viral receptor through the receptor-binding domain (RBD), whereas the S2 subunit promotes virus–cell membrane fusion [6,7]. According to the vital roles of S protein, current US FDA-approved vaccines focus on challenging immune responses against the S1 subunit of SARS-CoV-2 spike. Trimeric spike proteins in native form and prefusion-stabilized conformation have been applied [8–10].

Regarding the rapid mutation of SARS-CoV-2, the protective antibodies induced by first-generation vaccines based on the Wuhan strain are remarkably reduced. This evidence has generated awareness of vaccine efficiency following the report of the Delta variant, which causes severe symptoms [11,12]. The neutralizing epitopes on critical amino acids in the RBD are crucial in protecting against COVID-19 variants with relative mutations [13,14]. The mutations in the spike protein may impair the efficacy of neutralizing antibodies by modifying the protein binding activity with ACE2 [15]. Although mutated amino acids of Delta variants reside in the S1 domain, they do not participate in the contact surface between the RBD and ACE2. This evidence supports the possibility that the neutralizing antibodies may not be necessary to target the epitopes on the interaction surface of RBD [16,17].

The subregion of S1, called receptor-binding motif (RBM), consists of amino acids from 424-494. The participation of RBM in supporting the conformation of the RBD to form a complex with ACE2 was clarified regarding the crystal structure [18]. Although this region is indirectly involved in ACE2 binding, the specific mAbs that interact with this motif can interfere with the infectivity of SARS-CoV-2. Several SARS-CoV-2 RBD-specific mAbs have been developed by several companies. Some of them have been approved for emergency use in the treatment of COVID-19, such as bamlanivimab as a monotherapy, and the casirivimab/imdevimab cocktail, which blocks viral entry into human cells [19,20]. Nowadays, there are a number of mAbs in late-stage clinical studies or marketed for COVID-19, such as Adintrevimab (ADG20), TY027, and Regdanvimab [21]. Recently, a discontinuous epitope recognized by a neutralizing antibody, H014, was reported. Interestingly, the reactive amino acids, i.e., F374, F377, C379, G413, and W436, are conserved in Delta and accessible when the RBD is in the open state [22]. The enhancement of the binding affinity of RBD-ACE2 was reported in the W436R mutant [23]. In addition, W436R mutation contributes to more stabilization of the RBD folding structure [24]. The cross-neutralizing antibodies against W436R were observed in a convalescent patient [25]. A mouse mAb clone, 4051-MM43, showed cross-reactivity and neutralizing activity against wild-type (WT) SARS-CoV-2 and the Delta variants [26,27]. Regarding its property, it is worth characterizing the motif recognized by 4051-MM43 mAb.

In this study, the levels of neutralizing antibodies were determined in HCWs receiving a booster dose of BNT162b2 (Pfizer–BioNTech) after two doses of CoronaVac. The NeutraLISA was applied to determine overall neutralizing antibodies, which interfere with the complex formation of ACE2 and RBD. In addition, in-house competitive ELISA was established to determine the neutralizing antibodies targeting the epitope recognized by 40591-MM43 mAb. An amino acid participating in the epitope recognized by 40591-MM43 mAb was identified and its location was analyzed in the RBD conformational structure. The information obtained will provide an approach to improve the efficiency of the COVID-19 vaccine.

## 2. Materials and Methods

### 2.1. Subjects

This research was conducted in Chiang Mai, Thailand, from August to September 2021. HCWs from Maharaj Nakorn Chiang Mai Hospital provided written informed consent for blood samples and oral informed consent to participate in a longitudinal study of antibodies against the RBD of SARS-CoV-2 in plasma samples and survey responses. Fifty participants were enrolled in this study and visited between 10–11 August and 24–25 August 2021.

### 2.2. Ethical Policy

Ethical approval was received from the Ethics Committees at the Faculty of Associated Medical Sciences, Chiang Mai University (ethics approval number: AMSEC64EX-013) and the Research Ethics Committee, Faculty of Medicine, Chiang Mai University (ethics approval number: COM-2564-08458).

## 2.3. Sample Preparation and Design

A total of 10 mL of EDTA blood samples was collected once at 14 days post 3rd vaccination with BNT162b2. The EDTA blood samples were centrifuged at 3,000 rpm for 10 min at room temperature, and the plasma samples were collected and stored at $-20\ °C$ until analysis.

Plasma samples from the enrolled subjects were sent to the Associated Medical Sciences (AMS) Clinical Service Center at the Faculty of AMS, Chiang Mai University, to detect IgG antibodies against SARS-CoV-2. In samples, IgG antibodies against the RBD of SARS-CoV-2 were analyzed using the ARCHITECT i System (Abbott, Ireland) and compared with the in-house indirect enzyme-linked immunosorbent assay (ELISA). **In terms of neutralizing antibodies against the RBD of SARS-CoV-2, in-house** competitive ELISA was performed and compared with the SARS-CoV-2 NeutraLISA assay (EUROIMMUN, Germany, cat.no. EI2606-9601-4).

## 2.4. Statistics

The statistical analysis was conducted using GraphPad Prism version 8.2.1, GraphPad Prism (GraphPad Software, Inc.). Spearman and Pearson's analyses were performed to evaluate the correlation coefficient. The correlation between IgG antibodies anti-RBD versus percent inhibition of neutralizing antibodies by NeutraLISA and in-house competitive ELISA was determined using Spearman. The correlation analysis of in-house competitive ELISA and NeutraLISA was conducted by Pearson. For all statistical tests, two-tailed *p*-values less than 0.05 were considered statistically significant.

## 2.5. Detection of IgG Antibodies against the RBD of SARS-CoV-2

Antibody levels were determined using the ARCHITECT i System (Abbott, Ireland). Antibodies (IgG) to the SARS-CoV-2 spike protein RBD were detected in plasma using the Abbott assay. The number of antibodies in each examined sample was measured in units specific to the assay (AU/mL), according to the manufacturer's instructions with regards to the WHO standards.

## 2.6. Determination of Neutralizing Antibodies against the RBD of SARS-CoV-2 by SARS-CoV-2 NeutraLISA

Neutralizing antibodies against SARS-CoV-2 were determined using the SARS-CoV-2 NeutraLISA assay. The recombinant S1/RBD domain of the S protein of SARS-CoV-2 was coated on the plates, and the assay procedure was performed as described by EUROIMMUN. In brief, the controls and samples were diluted in a sample buffer containing soluble biotinylated ACE2 before being incubated in the reagent wells. If neutralizing antibodies were present in the sample, they competed with the ACE2 receptor for the binding sites of the SARS-CoV-2 S1/RBD proteins. In a second washing phase, unbound ACE2 was eliminated. A second incubation step with peroxidase-labeled streptavidin was performed to assess the bound biotinylated ACE2, which catalyzed a color reaction in the third reaction step. The intensity of the color produced was inversely proportional to the concentration of neutralizing antibodies in the sample.

## 2.7. Detection of the Antibodies Targeting the Epitope Recognized by mAb Using Competitive ELISA

The competitive ELISA was achieved by the competition of neutralizing antibodies against the RBD in vaccinated plasma and the spike neutralizing mouse mAb. The wells were coated with 50 μL of 1.0 μg/mL SARS-CoV-2 spike RBD-His recombinant protein (Sino Biological, Beijing, China) (Cat. 40592-V08H31) (diluted in coating buffer 1M NaHCO$_3$ (pH 9.6)) and kept overnight at 4 °C in the moisture chamber. The coated wells were washed four times with 0.05% Tween 20 in PBS (pH 7.4) (PBST) and non-specific binding was blocked with 200 μL of 2% skimmed milk in PBS at room temperature for 1 h. After washing, plasma samples at dilution 1:5 were combined with 2.5 μg/mL spike neutralizing

antibody mAb (40591-MM43) (Sino Biological, Beijing, China) (Cat. 40591-MM43), then added to ELISA wells and incubated for 1 h. The wells were washed and HRP-conjugated goat anti-mouse Igs was added at dilution 1:3,000 (KPL, Gaithersburg, MD, USA) (Cat. 074-1807). After incubation for 1 h, the wells were washed and 50 μL of TMB substrate was added (Seracare, Milford, MA, USA) (Cat. 5120-0076). The reaction was stopped with 1 N HCl, and the OD at 450 nm was measured.

### 2.8. In-house Indirect ELISA

For indirect ELISA, a 96-well microtiter plate (Greiner Bio-One) (Cat. 762071) was coated overnight at 4 °C with 50 μL per well of 1 μg/mL of SARS-CoV-2 spike RBD-His recombinant protein, WT and Delta (Sino Biological, Beijing, China) (Cat. 40592-V08H115) and W436R mutants (Sino Biological, Beijing, China) (Cat. 40592-V08H9), in coating buffer (pH 9.6) for detection of SARS-CoV-2 spike RBD antibodies. After blocking with 2% skimmed milk at 25 °C for 1 h, 1.0 μg/mL spike neutralizing antibody mAb (40591-MM43) with 2% skimmed milk in PBS was added to the coated wells. The plate was incubated at room temperature for 1 h and washed with washing buffer (PBST). After washing, HRP-conjugated goat anti-mouse Igs was added to each well for 1 h. After washing, TMB Substrate Solution was added and incubated for 10 min. The reaction was stopped by 1N HCl, and the OD at 450 nm was measured using a micro-plate reader (Hercuvan Lab Systems, Milton Cambridge, UK).

### 2.9. Western Blot Analysis

Purified SARS-CoV-2 spike RBD-His recombinant protein, WT and mutants (Delta and W436R) (50 μg/mL), was subjected to SDS-PAGE under reducing conditions, and then transferred to nitrocellulose membranes. The membrane was blocked with 2% skimmed milk in PBS. The SARS-CoV-2 (2019-nCoV) spike neutralizing mouse mAb (40591-MM43) (Sino Biological, Beijing, China) (Cat. 40591-MM43), SARS-CoV-2 (2019-nCoV) spike RBD mouse polyclonal antibody (pAb) (Sino Biological, Beijing, China) (Cat. 40592-MP01) or anti-6x His mAb (0.5 μg/mL, Invitrogen, Rockford, IL, USA) (Cat. MA1-21315) were separately added and incubated for 1 h at room temperature with shaking. After washing, the membranes were incubated with an HRP-conjugated goat anti-mouse immunoglobulin antibody (dilution 1:5,000 in blocking solution) for 1 h. The membranes were washed, and bands were visualized using KPL LumiGLO® Chemiluminescent Substrate System (Seracare, Milford, MA, USA) (Cat. 5430-0040). The resulting reactive protein bands were visualized using a ChemiDoc™ MP Imaging System (Bio-Rad, Hercules, CA, USA).

### 2.10. Structure Analysis of SAR-CoV-2 Spike Protein and ACE2

The amino acid residues participating in the complex structure of RBD with the host receptor ACE2 (PDB ID:6m0j) were analyzed using Prodigy [28]. The relative solvent accessibility value of amino acid residues in the RBD of SAR-CoV-2 S protein was analyzed using MISSENSE 3D [29] in RBD close (PDB ID: 7DDD) and open (PDB ID:7DDN) states. The amino acid residues of interaction between the SARS-CoV-2 spike RBD and ACE2, and RBD close and open state in the SAR-CoV-2-S protein, were labeled using UCSF Chimera software [30].

## 3. Results
### 3.1. Detection of IgG Antibodies against the RBD of SARS-CoV-2 Using ARCHITECT i System

Plasma samples from Thai HCWs at 14 days post third vaccination with BNT162b2 were examined for the levels of IgG antibodies against the RBD of SARS-CoV-2 by ARCHITECT i System (Abbott, Ireland) (n = 50). The antibody levels were classified into four groups (Figure 1).

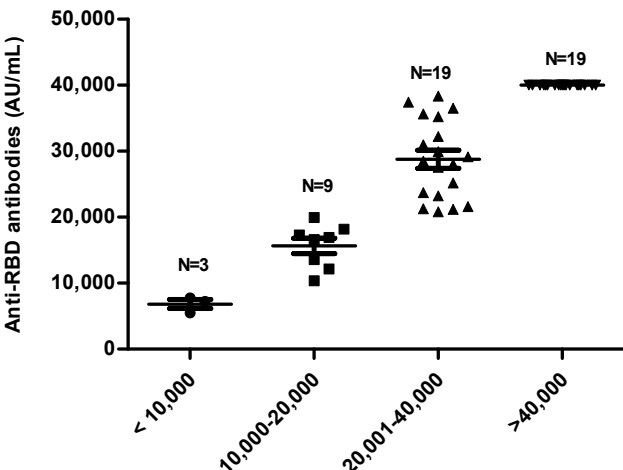

**Figure 1.** IgG antibodies against the RBD of SARS-CoV-2 levels in plasma samples, collected at 14 days post 3rd vaccination with BNT162b2 (*n* = 50), were measured by ARCHITECT i System (Abbott, Ireland).

*3.2. Validation of Neutralizing Antibodies against the RBD of SARS-CoV-2*

The competitive ELISA is used to identify a population of antibodies that recognize a specific motif on an RBD interacting with mAb 40591-MM43, with neutralizing activity against the WT and Delta variants. Competitive ELISA was employed to determine neutralizing antibodies against the RBD of SARS-CoV-2. Two assays were used to determine neutralizing antibodies in plasma: SARS-CoV-2 in-house competitive ELISA and NeutraLISA. The different levels of neutralizing antibodies between individuals are shown in Figure 2.

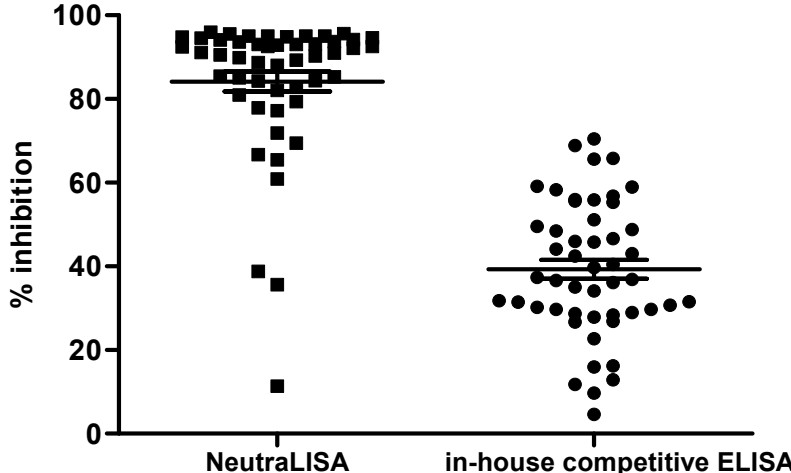

**Figure 2.** Comparison of the neutralizing antibodies against the RBD of SARS-CoV-2 in plasma samples collected at 14 days post 3rd vaccination with BNT162b2 by NeutraLISA and in-house competitive ELISA (*n* = 50).

The correlation between IgG antibodies anti-RBD (AU/mL) and percent inhibition of neutralizing antibodies by NeutraLISA and in-house competitive ELISA was determined using Spearman's correlation coefficients. Spearman's correlation analysis revealed a very strong positive, statistically significant association between antibodies and RBD detected by Architect i SARS-CoV-2 IgG II Quant with NeutraLISA ($r_s$ = 0.8576, *p* < 0.0001, Figure 3A), and with in-house competitive assay ($r_s$ = 0.8246, *p* < 0.0001, Figure 3B). However, $r_s$ from Spearman's correlation was significantly lower when the population with less than 25,000 AU/mL IgG antibodies anti-RBD was analyzed. Spearman's correlation analysis

demonstrated the moderate positivity between antibodies and RBD detected by Architect i SARS-CoV-2 IgG II Quant with NeutraLISA ($r_s$ = 0.5500, *p* = 0.0337), and with in-house competitive assay ($r_s$ = 0.5594, *p* = 0.0301).

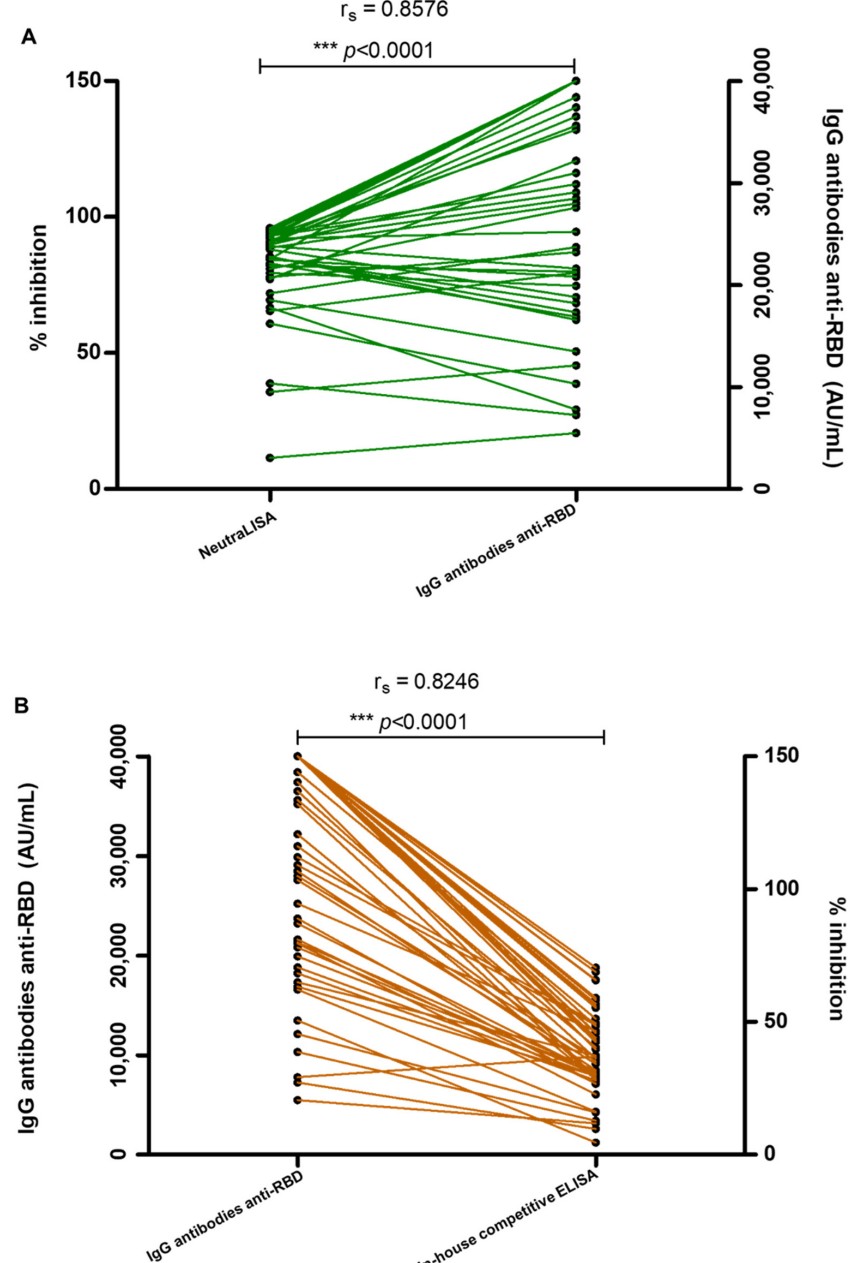

**Figure 3.** The comparison between IgG antibodies anti-RBD (AU/mL) and percent inhibition of neutralizing antibodies by NeutraLISA (**A**) and in-house competitive ELISA (**B**). The correlation between IgG antibodies anti-RBD (AU/mL) and percent inhibition of neutralizing antibodies by NeutraLISA and in-house competitive ELISA was performed using Spearman's correlation coefficients. Statistical significance was calculated using the two-tailed test (*** *p* < 0.0001).

The correlation analysis of in-house competitive ELISA and NeutraLISA was determined by Pearson's correlation coefficients. The *moderate* correlation between in-house and NeutraLISA was significant, as shown in Figure 4 (Pearson r = 0.6877; *p* < 0.0001; $R^2$ = 0.4729).

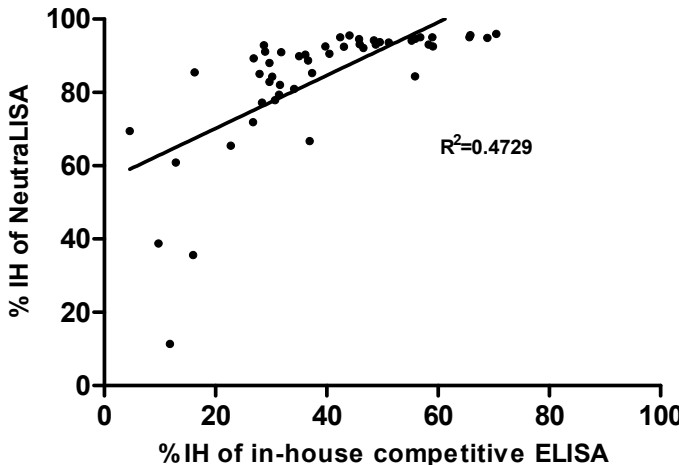

**Figure 4.** Correlation between the neutralizing antibodies against the RBD of SARS-CoV-2 by NeutraLISA and in-house competitive ELISA was analyzed using Pearson's correlation coefficients. Statistical significance was calculated using the two-tailed test ($n$ = 50).

*3.3. Binding Activity of 40591-MM43 mAb against WT and Mutants of SARS-CoV-2 Spike RBD Using Indirect ELISA*

To analyze the binding activity of 40591-MM43 mAb to SARS-CoV-2 spike RBD (WT, Delta, and W436R) protein, indirect ELISA was performed. SARS-CoV-2 spike RBD (WT, Delta, and W436R) was detected by spike neutralizing antibody mAb (40591-MM43). High optical density was obtained from WT and Delta, whereas the signal OD from W436R was decreased (Figure 5A). These data suggest that the mAbs that neutralize SARS-CoV-2 may bind a conserved epitope on the spike RBD. *The binding activity of 40591-MM43 mAb against each of the three RBDs (WT, Delta, and W436R) with different scalar concentrations of mAb (0.1, 0.5, 1, 2.5, and 5 µg/mL) is shown in Figure 5B. The OD obtained from WT and Delta reactions was in keeping with the concentration of mAb; however, the OD from W436R was remarkably lower, and not dependent on the concentration of mAb.*

**A**

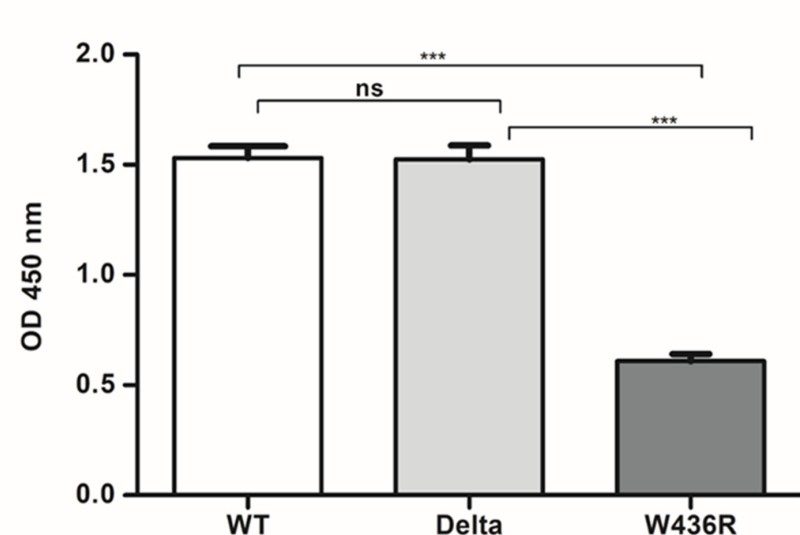

**Figure 5.** *Cont.*

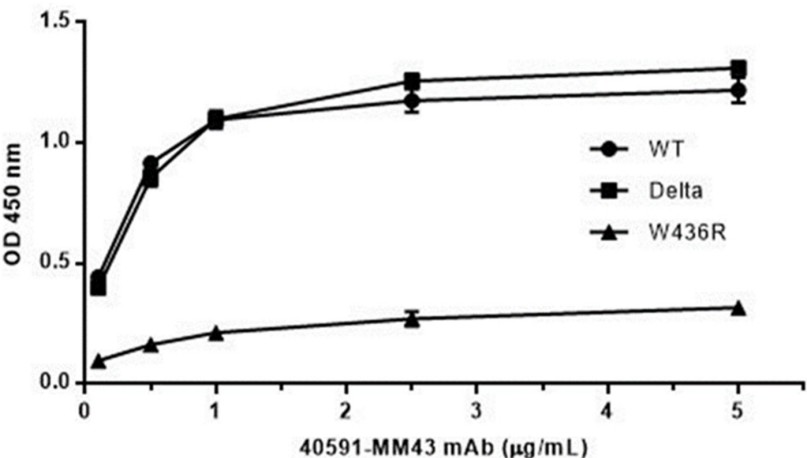

**Figure 5.** Binding activity of 40591-MM43 mAb to SARS-CoV-2 spike RBD WT, Delta, and W436R (**A**). The experiment is performed in triplicate. Statistical significance was determined by a two-tailed Student's test (*** $p < 0.0001$). Binding activity of 40591-MM43 mAb to SARS-CoV-2 spike RBD WT, Delta, and W436R with different scalar concentrations of mAb (**B**). The experiment is performed in triplicate.

### 3.4. Locating Amino Acid Residues on SAR-CoV-2 Spike Protein Structure

The interaction of the spike's RBD domain with the host ACE2 receptor (PDB ID:6m0j) is shown in Figure 6. The critical mutations in the Delta RBD are depicted in the crystal structure (Figure 6). The mutations in Delta, such as lysine at position 417 to asparagine (K417N), leucine at position 452 to arginine (L452R), threonine at position 478 to lysine (T478K), and glutamic acid at position 484 to lysine (E484K), are shown in green, whereas tryptophan at 436 (W436) is labeled in red. Relying on the SARS-CoV-2-S protein structure (PDB ID: 7DDD and 7DDN), the result showed that W436 is located at the RBM of the RBD domain (Figure 7). Focusing on W436, the relative solvent accessibility (RSA) is 3.5% in the close state of the RBD. In contrast, the RSA of W436 in the open state of the RBD is 35.2% (Figure 7).

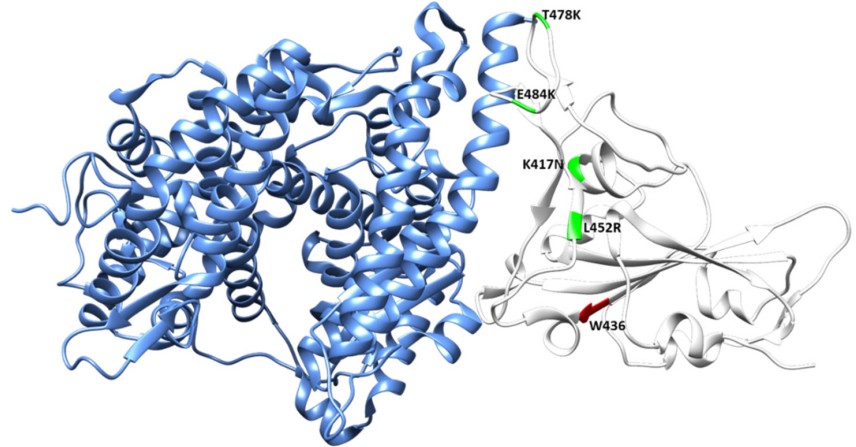

**Figure 6.** Interaction between the SARS-CoV-2 spike RBD (white) and ACE2 (cornflower blue) as shown by the crystal structure of the RBD/ACE2 complex. The amino acid W436 is labeled in red, whereas the point mutations of the Delta variant (K417N, L452R, E484K, and T478K) are labeled in green.

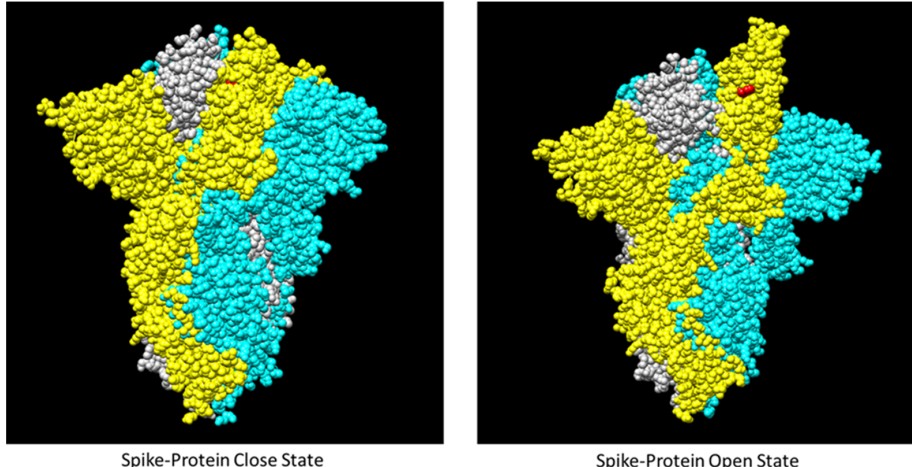

Spike-Protein Close State    Spike-Protein Open State

**Figure 7.** The RBD close and open states in the SARS-CoV-2-S protein (PDB ID: 7DDD and 7DDN). The chains A (yellow), B (cyan), C (white), and W436 residue (red) are displayed in the surface mode of UCSF Chimera.

*3.5. Determination of SARS-CoV-2 Spike RBD mAb to RBD WT and Mutants Interaction Using Western Blotting*

To determine the binding of SARS-CoV-2 spike RBD mAb clone 40591-MM43 to SARS-CoV-2 spike RBD-His recombinant protein (WT, Delta, and W436R), a Western blot analysis was conducted. The results demonstrated that mAb clone 40591-MM43 could not recognize the epitope on WT, Delta, and W436R (Figure 8, strips no. 1, 4, and 7); whereas, SARS-CoV-2 spike RBD mouse pAb (40592-MP01) (Figure 8, strips no. 2, 5, and 8) and anti-6x His mAb (Figure 8, strips no. 3, 6, and 9) bound to WT and Delta and W436R. This suggested that the 40591-MM43 that neutralizes SARS-CoV-2 binds to a conformational epitope on the spike RBD.

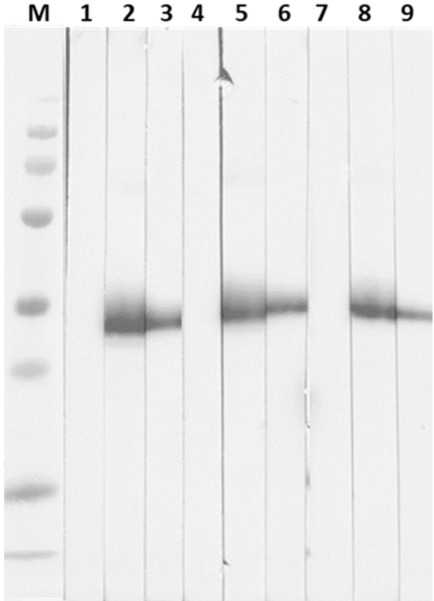

**Figure 8.** Western blot analysis of 40591-MM43 mAb to spike RBD-His recombinant protein (WT, Delta, and W436R). Proteins (strips 1–3 WT; strips 4–6 Delta; strips 7–9 W436R) on the blot were probed with SARS-CoV-2 spike RBD mAb (40591-MM43) (strips 1, 4, and 7); SARS-CoV-2 spike RBD mouse pAb (4059 2-MP01) (strips 2, 5, and 8); mouse anti-6x His mAb (strips no. 3, 6, and 9) followed by goat anti-mouse Igs conjugated with HRP.

## 4. Discussion

Several mAbs have been established for validating the neutralizing epitopes on S protein. Humanized and fully human mAbs have been applied in clinical studies. Currently, only two of them, i.e., casirivimab against the open state of RBD and imdevimab against the RBD core, are successfully approved by the FDA for passive immunotherapy in patients with moderate symptoms caused by the latter SARS-CoV-2 Delta variant [31,32]. Interestingly, the interactive sites of these mAbs are located in the RBM. They are tolerant to amino acid mutation of the Delta variant, although they non-competitively bind to epitopes of the SARS-CoV-2 protein RBD [20,33]. Considering the interaction between casirivimab and imdevimab, the B-cell receptors cannot efficiently access their epitopes while RBD is in the close stage. This complies with the phenomenon that the population receiving CoronaVac only cannot promote cross-protective antibodies, since the open state of the RBD is minor in S trimers. Modification of the spike of BNT162b2, using the HexaPro strategy, provides a greater possibility of retaining the RBD in the open state [34,35]. This results in better protection of the Delta variant in BNT162b2 vaccinated individuals [36].

Beyond the pandemic of the Delta variant in 2021, this also initiates concern around the currently available vaccine, with regard to the S protein of the Wuhan strain. The cross-protectivity of vaccinated individuals is remarkably reduced across all vaccine regimens, due to several amino acid mutations. In Thailand, the vaccine regimen for HCWs was two doses of CoronaVac, followed by a booster dose of BNT162b2. The comparative antibody titer of pre- versus post-BNT162b2 was reported in our former study [37]. Although the antibody titers are higher in post-BNT162b2, certain populations achieved low-to-moderate responding degree (Figure 1). Thus, it is interesting to verify the property of neutralizing antibodies against the Delta variant.

Since 40591-MM43 mAb harbors neutralizing activity against the Delta variant [27], it is worth comparing the relationships in the specific population of neutralizing antibodies activated in this study cohort. Sera from 50 volunteers were collected after 14 days post 3rd vaccination with BNT162b2, to analyze their neutralizing antibody characteristics with an in-house competitive ELISA, compared with the commercial assay, i.e., SARS-CoV-2 NeutraLISA. The in-house competitive ELISA aims to identify the population of human antibodies that recognize the specific motif on the RBD targeted by mAb 40591-MM43. In contrast, the NeutraLISA detects the overall population of antibodies interfering with the reactivity of the RBD and ACE2. Formerly, the anti-SARS-CoV-2 spike and RBD IgG titer in patients infected in Italy and Germany was reported to strongly correlate with the neutralizing antibody titers [38,39]. This evidence was also clearly demonstrated in this cohort using NeutraLISA and in-house competitive ELISA (Figure 3). Noticeably, the levels of neutralizing antibodies differ among individuals, especially in the moderate antibody titers. When reanalyzed using the samples with titers below 25,000 AU/mL, the correlation ($R^2$) values of NeutraLISA and in-house competitive ELISA were 0.5026 ($p = 0.0031$) and 0.3246 ($p = 0.0266$), respectively. Regarding Pearson's statistical analysis of in-house ELISA and NeutraLISA (r = 0.6877; $p < 0.0001$), the correlation is significant to a moderate degree. Since the $R^2$ (0.4729) is below 0.5, the strength of the model is rather low (Figure 4). Accordingly, the antibody response after boosting with BNT162b2 (Pfizer–BioNTech) probably does not effectively recognize the epitope of 40591-MM43 mAb. In addition, the antibodies specific to the 40591-MM43 mAb-related epitope did not correspond to the antibody titers determined by Architect i SARS-CoV-2 IgG II Quant. This evidence suggests that the vaccination regimen for this cohort is not sufficient to promote the antibodies that cross-protect the infection of WT and Delta variants.

The binding activity of 40591-MM43 mAb against WT and mutants, i.e., Delta and W436R, was further evaluated. Although it recognized the conserved motif of WT and Delta RBD, its immunoreactivity against the W436R mutant was markedly diminished (Figure 5). This information suggests that the W436 participates in the antigenic determinant of the RBD. This crucial amino acid is conserved among WT and Delta variants, and resides in the RBM compartment of the RBD. This supports the fact that the key residues, which

trigger the neutralizing antibodies, are not necessary to locate on the RBD-ACE2 contact surface [17,40].

The SARS-CoV-2-S protein contains both open and close structures found in the RBD. The open state is required for the S protein to bind to the ACE2 receptor. At least one RBD in the trimeric spike is positioned open, then turned outwards from S2, revealing the surface that attaches to the receptor. Regarding the analysis of the molecular structure of the SARS-CoV-2 spike (PDB ID: 7DDD and 7DDN), W436 is accessible while the RBD structure is in the open state. Regarding 3,854 prefusion trimers, the structure of the fully closed trimer and trimers with one RBD open represented approximately 31% and 55%, respectively [41]. Thus, the possibility to challenge the B-cell receptor against this occluded epitope consisting of W436 is substantially low. Although BNT162b2 has promoted the open conformation of RBD, a single booster dose is insufficient to trigger the antibody against this cross-protective epitope relating to 40591-MM43 mAb. This reflects low protection efficiency against the Delta variant. Accordingly, further booster with BNT162b2 should be suggested [42–44].

The Western blotting analysis demonstrated that 40591-MM43 mAb requires conformation structure on the RBD of WT, Delta and W436R to form the epitope. Thus, the synthetic peptide strategy is not applicable to vaccine development. To acquire B-cell activation against this cross-neutralizing epitope, the surface of the RBD should be accessible in every dimension. Recently, the recombinant RBD subunit, instead of the whole S protein, has been proposed for SARS-CoV-2 vaccines. The RBD vaccination strategy demonstrated a significant impact on eliciting elevated cross-neutralizing antibody titers, with subsequent protection against viral variants and the potential to limit community transmission [45]. Zhang et al. (2022) investigated the effect of adjuvants on the immunogenicity of three COVID-19 vaccine candidates, based on proteins, including RBD-Fc, RBD, and S-trimer [46]. The feasibility of inducing protective immunity against Omicron and future emerging sarbecoviruses relying on RBD was recently reported by Lui et al. (2022) [47]. This evidence complies with our findings, which deliberate the approach of RBD vaccine competency.

## 5. Conclusions

The suggested vaccine protocol for HCWs in Thailand in the years 2020–2021 was two doses of CoronaVac and a booster dose of BNT162b2. Since then, several studies have reported the efficacy in preventing the infection. The anti-RBD antibody level after the BNT162b2 booster is strongly associated with an overall neutralizing antibody against the Wuhan strain. However, protection against the Delta variant is not adequate, which has delivered huge concern for future vaccine design and vaccine regimens. Our study demonstrated that the specific population of antibodies recognizing a cross-neutralizing epitope on the RBM of the Delta variant is inadequate. This unique epitope is in the RBM and is not accessible when the conformation of the RBD is in a close state. To increase the possibility for B-cell receptors to access such an epitope, the RBD should be considered in acquiring the broad protection of SARS-CoV-2 variants.

**Author Contributions:** Conceptualization, C.T.; methodology, C.T. and W.T.; investigation, C.T., W.T. and K.T.; resources, C.T.; writing—original draft preparation, C.T., W.T. and K.T.; writing—review and editing, C.T., W.T. and K.T.; supervision, C.T.; funding acquisition, C.T. All authors have read and agreed to the published version of the manuscript.

**Funding:** This research work was supported by the Distinguished Research Professor Grant (NRCT 808/2563) of the National Research Council of Thailand, the Office of National Higher Education Science Research and Innovation Policy Council (NXPO), Thailand, through Program Management Unit for Competitiveness (PMU C), contract number C10F630145, Chiang Mai University, Thailand. Information Center For research and Socially Engagement Scholarship (Grant No. R000030259), Chiang Mai University, Thailand.

**Institutional Review Board Statement:** Not applicable.

**Informed Consent Statement:** Informed consent was obtained from all subjects involved in the study.

**Data Availability Statement:** The datasets used and/or analyzed during the current study are available from the corresponding author.

**Acknowledgments:** The authors are deeply grateful to all the involved in the study and would like to acknowledge the efforts of physicians, and support staff involved in the sample collection at the HCWs from Maharaj Nakorn Chiang Mai Hospital, Chiang Mai University. We also thank all study participants for their generous participation and contribution to this work.

**Conflicts of Interest:** The authors declare no conflict of interest.

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
