# Peer review of "The Occluded Epitope Residing in Spike Receptor-Binding Motif Is Essential for Cross-Neutralization of SARS-CoV-2 Delta Variant"

_cimb, doi:10.3390/cimb44070195_

Round 1

Reviewer 1 Report

The authors analyse the reactivity of BNT162b2 vaccinated HCW sera towards RBD and focus their attention on neutralizing antibodies.

The paper could be interesting but it is not completely clear which is the aim of the work.

1) First of all, in the introduction authors state that RBM (S424-494) is not involved in ACE binding. This is not completely exact. See, for example, the paper below....

Shang, J., Ye, G., Shi, K. et al. Structural basis of receptor recognition by SARS-CoV-2. Nature 581, 221–224 (2020). https://doi.org/10.1038/s41586-020-2179-y

2) The binding of mAb to RBD proteins (Fig. 5) should be repeated using a binding curve with different scalar concentrations of mAb.

3) It is not clear how authors decide to evaluate W436R mutation. In the introduction the authors state that RBM correspond to Spike 424-494. In the paragraph 2.4, they list some mutations and indicate that W436R is located in the RBM. In my humble opinion also L452R, T478K, E484K are located in the RBM region. Why authors decide to analyse only W436R?

Please clarify this point and make it more clear in the introduction

4) The in house competitive ELISA does not measure neutralizing antibodies. There is no ACE2 in this assay. The test measure the interference of serum Ab in the binding of mAb 40591-MM43 to RBD. It is completely different. Please modify the contribute of this assay in the whole story.

5) The main point to be clarified is the following. OK, mAb binding to RBD is markedly reduced when W436 is modified into R and mAb binding is towards a conformational epitope (WB with denatured protein is negative).

Which is the relation of these results with the immune response observed in HCW after three doses vaccination cycle?

Author Response

Manuscript ID: cimb-1744678

Title: The occluded epitope residing in spike receptor-binding motif is essential for cross-neutralization of SARS-CoV-2 Delta variant

Dear Editor,

We would like to thank the editor for the opportunity to submit the revision. The reviewer comments and suggestions have been carefully considered and responded point-by-point. We hope that the quality of the revised manuscript is suitable for publishing in Current Issues in Molecular Biology.

Thank you for your consideration.

Yours sincerely,

Prof. Dr. Chatchai Tayapiwatana

On behalf of the corresponding author

Reviewer 2 Report

This article deals with the decline in the effectiveness of the vaccination regimen used in Thailand. The authors confirmed the reduced efficacy of the currently used vaccination regimens with 2 vaccines due to the decrease in induced cross-protection by the appearance of 4 critical mutations in the RBD region of the Delta variant S protein. The work should be published because it confirms the effectiveness of the key modification used in vaccines - the stiffening of the S protein in the open conformation.

 Comments and suggestions:

Please consider replacing the statement in the abstract:

- "Although irrelevant to the ACE2 contact surface of the SARS-CoV-2 protein receptor-binding domain (RBD), the interactive sites of specific mAbs are in the receptor-binding motif (RBM) and are tolerant to amino acid mutations of the Delta variant. " on the simpler "It has been confirmed that the key amino acid residues in epitopes that induce the formation of neutralizing antibodies do not have to be on the RBD-ACE2 contact surface and may be conformationally hidden"

- and the reference to a change in the vaccination regimen, eg "Single BNT162b2 booster, not is sufficient to generate an effective level of neutralizing antibodies against the Delta variant epitope, which suggests a second booster dose of BNT162b2 vaccine to increase protection against this variant "

 Figure 2 - it would be good to indicate on the graph the statistical significance between the results of these two measurement methods.

 Line 121 - an unnecessary word in brackets - "variants". More understandable would be: spike RBD protein variants (…).

 Line 156 - the notation "(WT and variants; Delta and W436R)," is incomprehensible.

 In Chapter 2.2 it would be good to mention that the competition ELISA is designed to identify a population of human antibodies recognizing a specific motif on an RBD recognized by mAb 40591-MM43 with neutralizing activity against the Delta variant. The reader learns about it only in the discussion.

The Original Images for Blots to the figure 8 in the main article adds nothing if it is not properly signed. Tracks should be signed.

Author Response

(The authors gave the same response as above.)

Reviewer 3 Report

In the present manuscript, Thongkum et al. investigate the presence of antibodies (including neutralizing) 14 days following a third booster vaccine dose in Tawain focusing on the epitope recognized by the 40591-MM43 mAb. While the paper has some interest, it looks as if it is written earlier in the pandemic which evolves at an extremely fast pace. Nevertheless, there are some points that could be improved.

In the main document explain S protein as Spike, as you do in the abstract.

Please update the list of monoclonal antibodies that are currently used or are in development for covid.

To facilitate the reader please fill in the manufacturer of the vaccines etc BNT162b2 (Pfizer–BioNTech).  

In figure 3 the way the graph is made collides with the 2 different y axis. NeutraLISA even though being close to the left y axis is measured in % inhibition. This kind of representation may confuse the reader.

In the introduction, it would be welcome for the uninitiated reader if you could briefly analyze the mutations studied in the manuscript. 

Since you stratify patients into groups based on the anti-RBD antibodies, and the limit of 20000 (not 25,000) is used as the limit between the second and third group it would be nice to see how each group's IgG anti-RBD antibodies correlate with the neutralizing Elisas (in different graphs or in the same graph using different colors). On this prism, the lost correlation in low-level antibodies would be better visualized.

Could you please specify why you state that the correlation between in-house and NeutraLISA is not statistically significant??

W436 is not a variant but a mutation.

It would be nice if in the introduction you could explain in detail why you chose to study the W436 mutation in regards to the 40591-MM43 mAb. To make the study more comprehensible it would be welcome if you could explain in detail your study design, in a step by step fashion.

Further details are needed regarding the healthy donors (how were they chosen, what was their medication background and their age).

Could you please elaborate on the following phrase "Although BNT162b2 has promoted the open conformation of RBD, a single booster dose is not sufficient to trigger the antibody against this cross-protective epitope relating to 40591-MM43 mAb".

Author Response

(The authors gave the same response as above.)

Round 2

Reviewer 1 Report

Ok, authors modified the paper according to reviewer's suggestion.

Author Response

Dear Reviewer,

Thank you.

Best,

Chatchai

Reviewer 3 Report

I would like to thank the authors for considering the reviewers' suggestions. The quality of the manuscript has been profoundly improved. However, i still have an unresolved issue regarding a statistical description and interpretation that might confuse the reader and might also be misleading.

I am referring to my previous comment/response regarding point 7: the correlation between in-house Elisa and Neutralisa. As far as i am concerned a p value below the threshold the authors have stated in the material and methods part is considered as statistical significant. If the authors do not specify the threshold, p values below 0.05 are considered as statistical significant. In this case, nothing is mentioned in the appropriate descritpion of statistics part of the manuscript. Nonetheless, a correlation with a statistical significance does not necessarily connotes a meaningful and important correlation. In this particular univariate analysis however, a coefficient of determination around 0.5, means that 50% of the variation in the outcome of the Neutralisa has been explained just by predicting the outcome using the in-house elisa, which sounds at least partially satisfactory. In order to rescue the readers from potential confusion, please explain in detail your perspective and complement the statistics part of the manuscript.

.

Author Response

Dear Editor,

The suggestion of reviewer #3 has been carefully considered and modified.

English usage and typos have been improved.

I appreciate your consideration.

Best,

Chatchai
